# Scheimpflug Tomographic Indices for Classifying Normal, Down Syndrome and Clinical Keratoconus in Pediatric Patients

**DOI:** 10.3390/diagnostics14171932

**Published:** 2024-09-02

**Authors:** Renato Souza Oliveira, João Quadrado Gil, Andreia Rosa, Maria João Quadrado, Mauro Campos

**Affiliations:** 1Instituto Brasileiro de Oftalmologia—IBOL, Rio de Janeiro 22250-145, Brazil; 2Faculty of Medicine, University of Coimbra, 3004-504 Coimbra, Portugal; joaomqgil@gmail.com (J.Q.G.); andreia.rosa@gmail.com (A.R.); mariaromeira@gmail.com (M.J.Q.); 3Department of Ophthalmology and Visual Sciences, Paulista School of Medicine, Federal University of São Paulo—UNIFESP, Botucatu Street, 822 Vila Clementino, São Paulo 04023-062, Brazil; mscampos@uol.com.br; 4Centro de Responsabilidade Integrado de Oftalmologia, Centro Hospitalar e Universitário de Coimbra, 3004-504 Coimbra, Portugal

**Keywords:** keratoconus, Down Syndrome, corneal tomography

## Abstract

The study aimed to evaluate the precision of different Pentacam indices in diagnosing keratoconus (KC) in pediatric patients with and without Down Syndrome (DS) and determine suitable cutoff values. This prospective multicenter cross-sectional study evaluated 216 eyes of 131 patients aged 6–18 years (mean age 12.5 ± 3.2 years) using Pentacam. Patients were categorized into four groups: KC, forme fruste keratoconus (FK), DS, and control, excluding DS patients with topographic KC. Receiver operating characteristic curves were generated to determine the optimal cutoff points and compare the accuracy in identifying KC and FK in patients with and without DS. In DS patients, corneal morphology resembled KC features. The most effective indices for distinguishing KC in DS patients were the average pachymetric progression index (AUC = 0.961), higher-order aberration of the anterior cornea (AUC = 0.953), anterior elevation (AUC = 0.946), posterior elevation (AUC = 0.947), index of vertical asymmetry (AUC = 0.943), and Belin/Ambrosio enhanced ectasia total derivation value (AUC = 0.941). None of the indices showed good accuracy for distinguishing FK in DS patients. The thresholds of these indices differed significantly from non-DS patients. The results highlighted the need for DS-specific cutoff values to avoid false-positive or false-negative diagnoses in this population.

## 1. Introduction

Keratoconus (KC) is characterized by gradual thinning and conical bulging of the cornea. Typically identified during puberty or early adulthood, KC often remains undetected in pediatric patients until the emergence of clear clinical signs or significant symptoms, including pronounced vision impairment. Pediatric KC is characterized by rapid advancement and a high likelihood of requiring keratoplasty [1,2].

Down Syndrome (DS) is the genetic disorder most commonly associated with KC, with an incidence of 0.5% and 15%, which is 10–300 times greater than that in the general population [3,4]. The susceptibility of patients with DS to KC may arise from the combination of collagen-related disorders, to which these patients are prone, and the habit of frequent eye rubbing. Additionally, a potential candidate gene for keratoconus is located on chromosome 21, which reinforces the chromosomal link between these two diseases [5,6,7,8].

A pivotal aspect of contemporary KC management is the early diagnosis of the disease, which is essential for early intervention and preventing a decline in visual acuity [9]. The Pentacam (Oculus, Wetzlar, Germany) is known to facilitate early diagnosis of KC. It employs a rotating Scheimpflug imaging system to assess both the anterior and posterior corneal surfaces and perform corneal aberrometry, provides a comprehensive pachymetry map, and incorporates advanced algorithms and combined indices specifically crafted for KC identification [10,11]. Nevertheless, most of the normative data used in Pentacam examinations have been derived from adult individuals without DS, primarily those aged >21 years. Thus, normative data for the pediatric population, especially for children with conditions such as DS, are lacking [12]. Consequently, the more effective diagnostic indices used for KC detection in this population, along with their cutoff points, have been extrapolated from the data provided by adult databases [13,14].

This study aimed to assess the accuracy of various Pentacam indices and compare their optimal cutoff points in diagnosing KC in pediatric patients with and without DS.

## 2. Materials and Methods

### 2.1. Participants and Procedures

This multicenter cross-sectional study was conducted at the Instituto Brasileiro de Oftalmologia, Rio de Janeiro, Brazil; the Federal University of São Paulo, Brazil; and the University of Coimbra, Portugal. This study adhered to the principles of the Declaration of Helsinki and was approved by the ethics committees of the participating institutions. Prior to study commencement, all participants and their legal guardians signed informed consent forms tailored to their age and level of comprehension.

The participants at all three centers were evaluated using the Oculus Pentacam HR by the same investigator. The scans were obtained in the automatic release mode with the patient’s pupils in the natural state under scotopic conditions. Quality assessment was performed automatically, and only good-quality images were recorded for analysis. The participants (age, 6–18 years) were categorized into four groups: keratoconus (KC), fruste keratoconus (FK), patients with DS (DS), and control (C).

To avoid methodological bias, the DS group only included patients with DS who did not show KC in clinical or topographic assessments. No DS patient included in this study had clinical or topographic evidence of KC. For diagnosing KC, we considered the following criteria: traditional slit-lamp keratoconus signs (such as Vogt’s striae); abnormal topographic parameters (a skewed asymmetric bow tie, central or inferior steepening, or a claw pattern on topography); maximum simulated keratometry (Kmax) > 47.2 D; and inferior-superior index (IS) > 1.4 D at 6 mm [15,16]. To prevent misdiagnosis, all examinations were evaluated by two corneal specialists. The normal contralateral eye (based on the absence of clinical and topographic signs of KC) of a patient who exhibited the KC criteria in only one eye was defined as showing FK [17]. The control group consisted of participants without DS who showed normal results in clinical examinations and met the normal topographic criteria. Participants with a history of ocular disease, ocular surgery, ocular trauma, corneal scarring, or contact lens use were excluded.

### 2.2. Instruments

The Pentacam HR captures multiple corneal images using a Scheimpflug slit-rotating camera and transforms the elevation profile into wavefront data using Zernike polynomials [18]. A default setting of 25 images/s was used. The investigated indices are listed below:

Internal anterior chamber depth (ACD); corneal volume (VOL); flat (K1) and steep (K2) keratometric readings; mean curvature cornea power within the central 3 mm circle (Km); Kmax; corneal asphericity (Q); inferior-superior index (IS, index of surface variance (ISV); index of vertical asymmetry (IVA); anterior elevation from the best-fit sphere (AE); posterior elevation from the best-fit sphere (PE); corneal thickness at the thinnest point (TP); average pachymetric progression index (PPI-Avg); maximum pachymetric progression index (PPI-Max); Ambrosio’s relational thickness maximum (ART-Max); Belin/Ambrosio enhanced ectasia total derivation value (BAD-D); total aberration (TOA), low aberration (LOA), and high-order aberration (HOA) of the entire, anterior, and posterior cornea expressed as root mean square (RMS) data; third-order Zernike polynomials of vertical coma (Z3, −1); third-order oblique trefoil (Z3, −3); and fourth-order spherical aberration, Z4, 0 (SA). All Zernike coefficients were calculated based on height data for a pupil diameter of 6.0 mm.

### 2.3. Statistical Analysis

A linear model with random effects was used to compare the mean indices and ages across groups or countries, considering potential dependencies between the left and right eyes of the same patient. Post hoc analyses, utilizing multiple comparisons with Bonferroni correction, were performed to identify mean differences among the ocular disease groups while maintaining an overall significance level. The normality of the data for the mixed linear model was confirmed using the Kolmogorov–Smirnov test. Deviation from normality, as per Gelman and Hill, did not bias estimates [19]. The Mann–Whitney *U* test was used to compare mean ages between countries due to the absence of normality in age distribution.

Receiver operating characteristic (ROC) curves were used to differentiate between the KC, FK, and other study groups. The area under the ROC curve (AUC) was used to evaluate the discriminatory ability, which indicated the ability of the test to accurately categorize eyes with and without the disease. A perfect test was represented by an area of 1.0, while a value of 0.5 indicated a test with no discriminatory ability [20]. Sensitivity and specificity were calculated for the optimum cutoff values derived from ROC curves based on the Youden index. Graphically, this point is the maximum vertical distance between the ROC curve and the equal diagonal line [21]. Statistical analyses were conducted using Stata/SE statistical software (version 14.0, 2015; Stata Corp, College Station, TX, USA), and *p*-values less than 0.05 were considered statistically significant.

## 3. Results

A total of 216 eyes of 131 patients (males, 55.1%) were evaluated in this study. The mean age of the patients was 12.5 ± 3.2 years. All groups, except the KC group, showed similar characteristics in terms of age and sex, with the KC group showing a higher average age than the other groups (14.6 vs. 12 years) and a substantial male predominance (80.3%) (Table 1).

Table 2 provides a comprehensive overview of the Pentacam parameters in the studied groups.

Apart from the anticipated differences in keratometric, aberrometric, and corneal thickness values based on the definition of each study group, the DS group notably exhibited a more curved (Km = 45.90) and aberrated (RMS HOA = 0.65) cornea than all the other groups, except the KC group. Corneal thickness was almost the same in the KC group (484 µm) and DS group (486 µm). For composite indices such as BAD-D and ART-Max, patients in the DS group displayed values more akin to those in the FK group but distinct from those in the KC group. Additionally, the DS group exhibited the lowest corneal volume among all the groups (55.61), including the KC group (58.35; *p* < 0.001).

Table 3 presents the cutoff values, sensitivity (SN), specificity (SP), and AUC values for the indices identified as the most effective in distinguishing eyes with KC in the DS and C groups, as well as the comparison between the FK and DS groups.

Additional data for all other analyzed parameters are provided in the Appendix A.

Table 4 illustrates the SN and SP when the cutoff values established for the control group were applied to the DS group. For comparison, the SN and SP for the specific cutoff points designed for the DS group are provided.

Figure 1 allows for the comparison of the ROC curves of the most effective indices for detecting KC in the DS group.

## 4. Discussion

Our data revealed that individuals with DS had corneas that were steeper, thinner, and more aberrant than those in the control group. Additionally, they displayed distinct values for the diagnostic indices of KC, such as BAD-D, ART-Max, and PPI, which were more similar to the values found in the FK group. These findings indicate that the corneal morphology in patients with DS often deviates from that in normal individuals, exhibiting characteristics typical of those who develop KC. These findings were consistent with those of previous studies [13,14,22,23]. Moreover, these results imply that the extrapolation of normative databases and recommended indices and cutoff points established in non-DS patients to this specific population will yield incorrect findings.

The diagnosis of KC in patients with DS may be challenging. Since the identification of KC depends on the adequate communication of symptoms, prompt referral, and cooperative participation in eye examinations, KC may remain undiagnosed in a substantial number of individuals with DS. Moreover, the distinct corneal structure of these patients may generate more false-positive findings or, even worse, more false-negative findings than in non-DS patients, especially if they are assessed using the same indices and cutoff points used for non-DS patients.

Our findings revealed that PPI-Avg exhibited the highest AUC for distinguishing KC in patients with DS, followed by AE, PE, BAD-D, IVA, total aberration, and HOA of the anterior corneal surface (Fig. 1). However, TP, VOL, ACD, and trefoil lacked adequate discriminative power for KC in patients with DS, rendering them unsuitable for inclusion in the screening analysis. Our analysis also suggests that a single index may be sufficient, since some of the indices listed above, especially PPI-Avg and HOA RMS, showed outstanding AUC values above 0.95.

Our study included only DS patients without topographic evidence of KC. It is known that KC in DS patients exhibits morphological characteristics similar to those in non-DS patients, with the primary distinction between these groups lying in the prevalence and severity of the disease rather than in the morphological characteristics of KC itself [3,24]. In fact, what makes distinguishing KC in the DS group more challenging is the distinct characteristics of the cornea in non-KC DS patients, which differ from those in non-DS patients [25,26]. Thus, although we did not directly compare DS patients with KC to those without KC, our study provides valid cutoff points for the studied indices. Our findings are supported by Asgari et al., who conducted a study comparing DS patients with and without KC and obtained results similar to ours [27]. Our data also revealed different optimal cutoff points for distinguishing KC in patients with and without DS. In fact, when examining the ROC curve of the KC group against the DS group and employing the cutoff values established in the KC group vs. group C, we observed a slight decrease in SN but a notable reduction in SP (Table 4). Thus, the failure to utilize specific cutoff points tailored to this population may lead to a substantial increase in false-positive diagnoses. Therefore, we propose that, when assessing children with DS, employing cutoff points specific to this population is crucial for achieving optimal sensitivity and specificity.

The KC-like corneal morphology observed in patients with DS may account for the significant decrease in the accuracy of certain indices in distinguishing KC within the DS group in comparison with the control group, especially TP (AUC, 0.919–0.524), Km (AUC, 0.895–0.532), and Q (AUC, 0.925–0.808). Conversely, combined indices such as ART-Max and BAD-D maintained excellent accuracy in discriminating KC in patients with DS (AUCs of 0.916 and 0.941, respectively). Moreover, the similarities in the curvature and thickness parameters between the DS and FK groups may explain the low AUC values found in the ROC curves for FK within the DS group.

A limitation of the present study is the absence of corneal biomechanics and corneal epithelial map assessments, which are advanced methods for KC detection. Indices such as the Corvis Biomechanical Index (CBI) and integrated Tomographic and Biomechanical Index (TBI), along with the epithelial thickness map parameters and patterns, have shown high accuracy in distinguishing eyes with mild KC from normal eyes [28,29,30]. These indices were not assessed in our study and can be the subject of future investigations.

In conclusion, pediatric patients with DS exhibit a unique corneal morphology, which may pose challenges in diagnosing KC and therefore necessitate meticulous corneal evaluation. Consequently, we propose the use of the following indices, along with their corresponding threshold values, for this specific population: PPI-AVG (>1.19), HOA RMS (>0.834), PE (>12), AE (>8), IVA (>0.25), and BAD-D (>2.71).

## Figures and Tables

**Figure 1 diagnostics-14-01932-f001:**
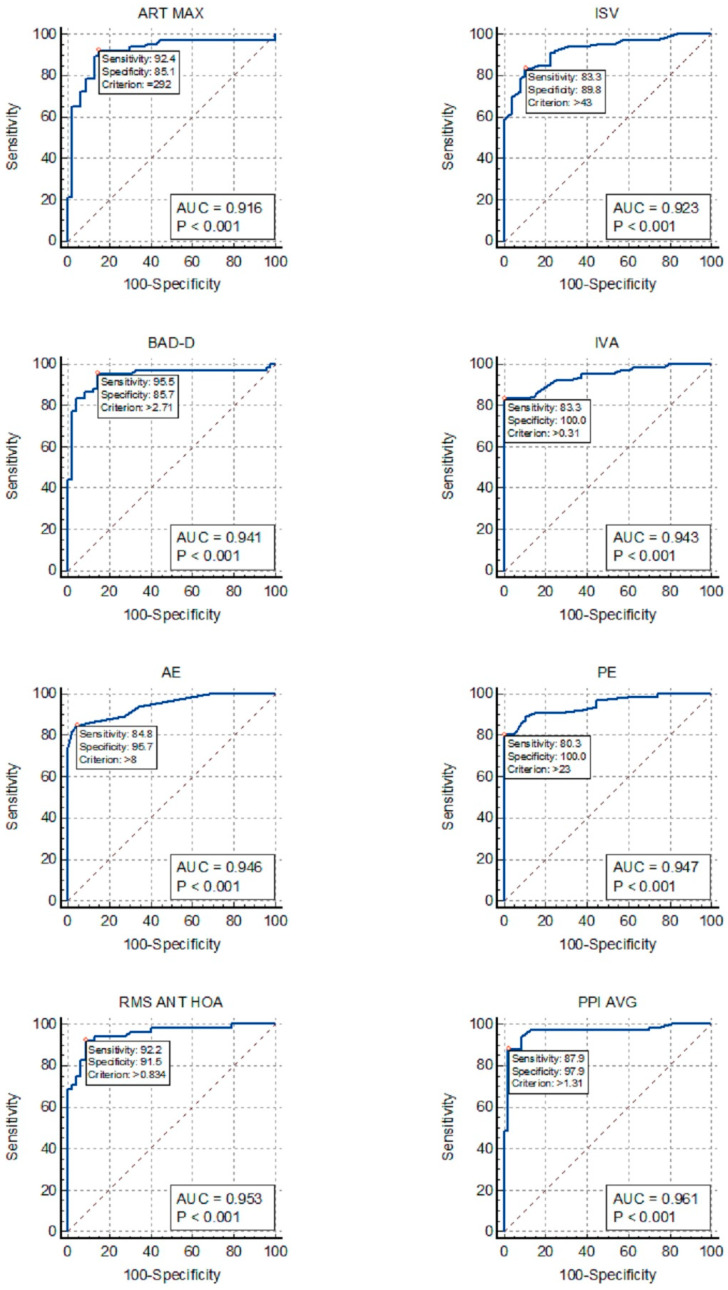
ROC curves for the detection of keratoconus in patients with Down Syndrome: ISV, index of surface variance; IVA, index of vertical asymmetry; BAD-D, Belin/Ambrosio enhanced ectasia total derivation value; PPI-AVG, average pachymetric progression index; ART-MAX, Ambrosio’s relational thickness maximum; RMS ANT-HOA, high-order aberration of the anterior cornea; AE, anterior elevation from the best-fit sphere; PE, posterior elevation from the best-fit sphere.

**Table 1 diagnostics-14-01932-t001:** Ocular characteristics in each group.

	Control (*n* = 87; 40.3%)	Keratoconus (*n* = 66; 30.5%)	Forme Fruste Keratoconus (*n* = 14; 6.5%)	Down Syndrome (*n* = 49; 22.7%)	Total (*n* = 216; 100.0%)	*p*
**Sex**						**<0.001 ^a^**
Male	49 (56.3)	53 (80.3)	9 (64.3)	32 (65.3)	199 (55.1)	
Female	38 (43.7)	13 (19.7)	5 (35.7)	17 (34.7)	162 (44.9)	
**Age (years)**						**<0.001 ^b^**
Mean ± SD	12.0 ± 2.9	14.6 ± 2.4	14.0 ± 2.5	12.5 ± 3.2	12.5 ± 3.1	
Median (IQR)	11.0 (10.0–14.0)	15.0 (12.0–16.0)	15.0 (11.8–16.0)	13.0 (9.5–15.5)	12.0 (10.0–15.0)	
**Age Range**						**<0.001 ^a^**
6–12 years	52 (59.8)	18 (27.3)	5 (35.7)	24 (49.0)	185 (51.2)	
13–18 years	35 (40.2)	48 (72.7)	9 (64.3)	25 (51.0)	176 (48.8)	

*p*-descriptive level of the chi-square test (a), and linear model with random effects (b). Interquartile range (IQR) (P25–P75). Bonferroni multiple comparison test: Control = astigmatism = Down Syndrome < keratoconus. SD: standard deviation.

**Table 2 diagnostics-14-01932-t002:** Summary measures of ocular indices in each group.

	Control	Keratoconus	Forme Fruste Keratoconus	Down Syndrome	Total	*p*
K1	42.65 ± 1.13 ^A^	44.99 ± 3.38 ^B^	42.29 ± 1.26 ^A^	45.01 ± 1.81 ^B^	43.21 ± 2.36	<0.001
K2	43.56 ± 1.20 ^A^	48.83 ± 4.55	43.89 ± 1.91 ^A^	46.89 ± 2.20 ^B^	45.91 ± 3.03	<0.001
K med	43.10 ± 1.16 ^A^	46.80 ± 3.80 ^B^	43.06 ± 1.53 ^A^	45.90 ± 1.88 ^B^	44.50 ± 2.54	<0.001
Cyl	0.91 ± 0.35 ^A^	3.84 ± 2.23 ^B^	1.50 ± 0.90 ^A^	1.89 ± 1.33 ^A^	2.70 ± 1.76	<0.001
Q	0.35 ± 0.13 ^A^	0.86 ± 0.40 ^B^	0.40 ± 0.17 ^A^	0.52 ± 0.14 ^A^	0.51 ± 0.27	<0.001
Kmax	44.08 ± 1.23 ^A^	53.95 ± 7.14 ^C^	44.79 ± 2.16 ^A^	47.66 ± 2.28 ^B^	47.34 ± 4.78	<0.001
TP	548.38 ± 31.63	484.03 ± 38.98 ^A^	519.64 ± 35.30	486.82 ± 34.28 ^A^	521.90 ± 43.55	<0.001
K1 post	6.05 ± 0.18 ^A^	6.52 ± 0.65 ^B^	5.98 ± 0.21 ^A^	6.24 ± 0.27 ^A^	6.14 ± 0.40	<0.001
K2 post	6.33 ± 0.20 ^A^	7.34 ± 0.83 ^B^	6.39 ± 0.34 ^A^	6.63 ± 0.32 ^A^	6.70 ± 0.54	<0.001
Cyl post	0.28 ± 0.11 ^A^	0.83 ± 0.44 ^B^	0.42 ± 0.18 ^A^	0.39 ± 0.23 ^A^	0.56 ± 0.32	<0.001
VOL	61.06 ± 3.25 ^B^	58.35 ± 3.33	58.74 ± 4.15 ^A^	55.61 ± 3.16 ^A^	59.61 ± 4.04	<0.001
ACD	3.50 ± 0.39	3.69 ± 0.41 ^B^	3.47 ± 0.44	3.47 ± 0.34	3.47 ± 0.43	0.001
AE	2.64 ± 1.18 ^A^	19.47 ± 11.54 ^C^	3.86 ± 1.75 ^B^	4.38 ± 2.58 ^B^	6.81 ± 8.02	<0.001
PE	5.07 ± 2.69 ^A^	40.92 ± 23.55 ^C^	7.64 ± 5.00 ^B^	7.74 ± 5.13 ^B^	13.08 ± 17.09	<0.001
PPI-Avg	0.94 ± 0.11 ^A^	1.81 ± 0.55 ^B^	1.05 ± 0.14 ^A^	0.96 ± 0.19 ^A^	1.13 ± 0.42	<0.001
PPI-Max	1.19 ± 0.16 ^A^	2.66 ± 0.93 ^B^	1.47 ± 0.29 ^A^	1.36 ± 0.41 ^A^	1.52 ± 0.71	<0.001
ART-Max	471.33 ± 79.52 ^C^	209.21 ± 93.61 ^A^	366.50 ± 78.70 ^B^	378.34 ± 94.57 ^B^	392.09 ± 125.91	<0.001
BAD-D	0.83 ± 0.48 ^A^	6.91 ± 3.92 ^C^	1.55 ± 0.90 ^B^	1.90 ± 1.06 ^B^	2.31 ± 2.84	<0.001
ISV	16.32 ± 4.26 ^A^	72.76 ± 34.65 ^C^	23.00 ± 5.55 ^A^	29.98 ± 9.17 ^B^	35.79 ± 24.70	<0.001
IVA	0.15 ± 0.14 ^A^	0.71 ± 0.39 ^B^	0.19 ± 0.07 ^A^	0.20 ± 0.07 ^A^	0.27 ± 0.28	<0.001
IS	0.38 ± 0.28 ^A^	4.08 ± 2.98 ^B^	1.03 ± 0.56 ^A^	0.74 ± 0.52 ^A^	1.24 ± 1.90	<0.001
RMS TOTAL	1.29 ± 0.32 ^A^	8.93 ± 5.39 ^C^	2.26 ± 0.71 ^A^	2.52 ± 1.09 ^B^	3.84 ± 3.51	<0.001
RMS LOA	1.23 ± 0.32 ^A^	8.65 ± 5.24 ^C^	2.19 ± 0.70 ^A^	2.42 ± 1.10 ^B^	3.74 ± 3.41	<0.001
RMS HOA	0.36 ± 0.10 ^A^	2.15 ± 1.33 ^B^	0.52 ± 0.17 ^A^	0.65 ± 0.21 ^A^	0.78 ± 0.88	<0.001
RMS TOTAL (anterior)	1.51 ± 0.32 ^A^	10.29 ± 6.12 ^C^	2.73 ± 0.91 ^A^	2.62 ± 1.09 ^A^	4.34 ± 4.03	<0.001
RMS LOA (anterior)	1.46 ± 0.32 ^A^	9.96 ± 5.94 ^C^	2.65 ± 0.89 ^A^	2.53 ± 1.09 ^A^	4.24 ± 3.91	<0.001
RMS HOA (anterior)	0.37 ± 0.08 ^A^	2.46 ± 1.50 ^B^	0.58 ± 0.23 ^A^	0.63 ± 0.24 ^A^	0.85 ± 1.02	<0.001
RMS TOTAL (posterior)	0.75 ± 0.12 ^A^	2.46 ± 1.33 ^B^	1.01 ± 0.29 ^A^	0.87 ± 0.23 ^A^	1.21 ± 0.84	<0.001
RMS LOA (posterior)	0.73 ± 0.12 ^A^	2.37 ± 1.29 ^B^	0.98 ± 0.29 ^A^	0.81 ± 0.21 ^A^	1.17 ± 0.82	<0.001
RMS HOA (posterior)	0.18 ± 0.03 ^A^	0.64 ± 0.34 ^B^	0.25 ± 0.08 ^A^	0.27 ± 0.12 ^A^	0.29 ± 0.23	<0.001
H COMA z3-1	0.13 ± 0.07 ^A^	0.92 ± 0.86 ^B^	0.18 ± 0.11 ^A^	0.22 ± 0.14 ^A^	0.31 ± 0.48	<0.001
V COMA z3-1	0.12 ± 0.08 ^A^	1.18 ± 0.95 ^B^	0.31 ± 0.25 ^A^	0.19 ± 0.15 ^A^	0.37 ± 0.57	<0.001
TREFOIL z3-3	0.08 ± 0.10 ^A^	0.32 ± 0.44 ^B^	0.07 ± 0.09 ^A^	0.17 ± 0.14 ^A^	0.14 ± 0.23	<0.001
SA	0.29 ± 0.17 ^A^	0.64 ± 0.88 ^B^	0.28 ± 0.14 ^A^	0.18 ± 0.11 ^A^	0.31 ± 0.43	<0.001

*p*-descriptive level of the linear model with random effects. (A), (B), and (C) show distinct means according to multiple comparisons with Bonferroni correction. ACD: anterior chamber depth; VOL: corneal volume; K1: flat corneal meridian; K2: steep corneal meridian; Km: mean curvature power of the cornea; Kmax: maximum simulated keratometry; Q: corneal asphericity; IS: inferior-superior index; ISV: index of surface variance; IVA: index of vertical asymmetry; AE: anterior elevation from the best-fit sphere; PE: posterior elevation from the best-fit sphere; TP: corneal thickness at the thinnest point; PPI-Avg: average pachymetric progression index; PPI-Max: maximum pachymetric progression index; ART-Max: Ambrosio’s relational thickness maximum; BAD-D: Belin/Ambrosio enhanced ectasia total derivation value; TOTAL: total aberration; LOA: low aberration; HOA: high-order aberration of the entire, anterior, and posterior cornea expressed as root mean square (RMS) data; H COMA Z3-1: third-order Zernike polynomials of horizontal coma; V COMA Z3-1: third-order Zernike polynomials of vertical coma; TREFOIL Z3-3: third-order oblique trefoil; SA: fourth-order spherical aberration.

**Table 3 diagnostics-14-01932-t003:** The values of the best indices for KC discrimination and their diagnostic accuracy.

	KC vs. Control Group				KC vs. DS Group				FK vs. DS Group			
	Cutoff	SN	SP	AUC	Cutoff	SN	SP	AUC	Cutoff	SN	SP	AUC
**AE**	4.5	97.0%	94.3%	0.991	8	84.8%	95.7%	0.946	6	27.7%	100%	0.557
**PE**	9.5	92.4%	92.0%	0.981	12	90.9%	85.1%	0.947	9	38.3%	78.6%	0.519
**PPI-AVG**	1.13	97.0%	95.4%	0.974	1.19	95.5%	89.4%	0.961	0.92	53.2%	92.9%	0.715
**ART-MAX**	358	97.0%	93.1%	0.972	292	92.4%	85.1%	0.916	398	38.3%	78.6%	0.549
**BAD**	1.74	97.0%	96.6%	0.982	2.71	95.5%	85.7%	0.941	0.95	89.8%	35.7%	0.587
**ISV**	23.5	97.0%	95.4%	0.997	34	90.9%	75.5%	0.923	22	77.6%	57.1%	0.732
**IVA**	0.22	93.9%	92.0%	0.965	0.25	92.4%	75.0%	0.943	0.18	60.4%	57.1%	0.542
**ANT-HOA**	0.55	98.0%	96.4%	0.998	0.83	92.2%	91.5%	0.953	0.45	83.0%	37.5%	0.606

KC, keratoconus; DS, Down Syndrome; FK, forme fruste keratoconus; SN: sensitivity; SP: specificity; AUC: area under the receiver operating characteristic curve; AE: anterior elevation from the best-fit sphere; PE: posterior elevation from the best-fit sphere; PPI-AVG: average pachymetric progression index; ART-MAX: Ambrosio’s relational thickness maximum; BAD: Belin/Ambrosio enhanced ectasia total derivation value; ISV: index of surface variance; IVA: index of vertical asymmetry; ANT-HOA: high-order aberration of the anterior cornea.

**Table 4 diagnostics-14-01932-t004:** Sensitivity and specificity of Pentacam indices in the DS group when applying cutoff values designed for the control group.

	KC vs. DS Group (Cutoff Values for the Control Group)			KC vs. DS Group (Specific Cutoff Values)			
	CUTOFF	SN	SP	CUTOFF	SN	SP	AUC
**AE**	4.5	93.9%	65.9%	8	84.8%	95.7%	0.946
**PE**	9.5	90.9%	74.4%	12	90.9%	85.1%	0.947
**PPI-Avg**	1.13	96.9%	87.2%	1.19	95.5%	89.4%	0.961
**ART-Max**	358	96.9%	51.0%	292	92.4%	85.1%	0.916
**BAD-D**	1.74	96.9%	59.2%	2.71	95.5%	85.7%	0.941
**IVA**	0.22	93.9%	62.5%	0.25	92.4%	75%	0.943
**ISV**	23.5	96.9%	38.8%	34	90.9%	75.5%	0.923
**ANT-HOA**	0.55	98.0%	46.8%	0.834	92.2%	91.5%	0.953

KC, keratoconus; DS, Down Syndrome; SN, sensitivity; SP, specificity; AUC, area under the receiver operating characteristic curve; AE, anterior elevation from the best-fit sphere; PE, posterior elevation from the best-fit sphere; PPI-AVG, average pachymetric progression index; ART-MAX, Ambrosio’s relational thickness maximum; BAD-D, Belin/Ambrosio enhanced ectasia total derivation value; IVA, index of vertical asymmetry; ISV, index of surface variance; ANT-HOA, high-order aberration of the anterior cornea.

## Data Availability

The datasets used and/or analyzed during the current study are available from the corresponding author on reasonable request.

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
