# Peer review of "Scheimpflug Tomographic Indices for Classifying Normal, Down Syndrome and Clinical Keratoconus in Pediatric Patients"

_diagnostics, 2024, doi:10.3390/diagnostics14171932_

Round 1

Reviewer 1 Report

Comments and Suggestions for Authors

The author aimed to identify the Pentacam index with the potential for detecting keratoconus in pediatric patients with Down syndrome. However, the study design compares “keratoconus with or without Down syndrome” against “Down syndrome without keratoconus”. Am I right? The author should define the patient groups more explicitly in the Materials and Methods section or consider revising the title. Moreover, I recommend the author show the ROC plots in addition to the table showing the cutoff points, sensitivity, and specificity.

Author Response

*The author aimed to identify the Pentacam index with the potential for detecting keratoconus in pediatric patients with Down syndrome. However, the study design compares “keratoconus with or without Down syndrome” against “Down syndrome without keratoconus”. Am I right? The author should define the patient groups more explicitly in the Materials and Methods section or consider revising the title. Moreover, I recommend the author show the ROC plots in addition to the table showing the cutoff points, sensitivity, and specificity.

Thank you for your thoughtful and detailed feedback. We appreciate your careful review of our manuscript and your valuable suggestions for improvement.

We chose to adjust the title to better align with the study's methodology. Accordingly, we have modified not only the title but also some sections of the abstract and introduction. These changes are highlighted in red. We believe that the methodology, objectives, and conclusions of the study are now more consistent with one another.

As suggested, we have included a figure with the ROC curves of the main indices.

We remain available to provide any further clarifications that may be needed. Thank you again for your consideration and collaboration.

Reviewer 2 Report

Comments and Suggestions for Authors

*The authors aimed to evaluate the precision of different Pentacam indices in diagnosing keratoconus in pediatric patients with Down syndrome and determine suitable cutoff values. The work was interesting and well-presented. However, I have a few concerns.

*In Table 1 and Table 2, you have reported only one p-value, all statistically meaningful. This shows that one group has different values among 4 enrolled groups, but it does not reveal the exact difference between different groups. So, I recommend providing a separate comparison between the DS group and others, each contains a distinct p-value.

*Regarding ACD in Table 2, is the p-value correct? Please check it since the values are so close in different groups.

Comments on the Quality of English Language

Minor editing.

Author Response

*In Table 1 and Table 2, you have reported only one p-value, all statistically meaningful. This shows that one group has different values among 4 enrolled groups, but it does not reveal the exact difference between different groups. So, I recommend providing a separate comparison between the DS group and others, each contains a distinct p-value.

Thank you for your thoughtful and detailed feedback. We appreciate your careful review of our manuscript and your valuable suggestions for improvement.

The p-value is presented as a global statistic. Distinct groups are later indicated using footnotes or symbols. It is not feasible to provide pairwise comparisons directly in the table, as four groups would result in six comparisons, which would require a separate table—something that is not typically done in this kind of paper.

In Table 1, for the comparison of means, the distinct groups are clearly indicated in the table’s footer. However, for percentage comparisons, it is not possible to mark them similarly. These observations are based on the standardized adjusted residuals, as detailed in the methodology.

In Table 2, distinct groups with statistically significant differences are marked with letters "A," "B," etc.

*Regarding ACD in Table 2, is the p-value correct? Please check it since the values are so close in different groups.

Regarding ACD, the p-value is correct—since the variability was small, it was possible to identify minor differences. We reviewed this index and found the same value previously reported.

We remain available to provide any further clarifications that may be needed. Thank you again for your consideration and collaboration.

Round 2

Reviewer 1 Report

Comments and Suggestions for Authors

The author appropriately responds to my comment.